# Altered Spatial Composition of the Immune Cell Repertoire in Association to CD34^+^ Blasts in Myelodysplastic Syndromes and Secondary Acute Myeloid Leukemia

**DOI:** 10.3390/cancers13020186

**Published:** 2021-01-07

**Authors:** Marcus Bauer, Christoforos Vaxevanis, Haifa Kathrin Al-Ali, Nadja Jaekel, Christin Le Hoa Naumann, Judith Schaffrath, Achim Rau, Barbara Seliger, Claudia Wickenhauser

**Affiliations:** 1Institute of Pathology, Martin Luther University Halle-Wittenberg, Magdeburger Str. 14, 06112 Halle, Germany; Marcus.bauer@uk-halle.de; 2Institute of Medical Immunology, Martin Luther University Halle-Wittenberg, 06112 Halle, Germany; christoforos.vaxevanis@uk-halle.de (C.V.); barbara.seliger@uk-halle.de (B.S.); 3Department of Hematology/Oncology, University Hospital Halle, 06112 Halle, Germany; haifa.al-ali@uk-halle.de (H.K.A.-A.); nadja.jaekel@uk-halle.de (N.J.); christin.naumann@uk-halle.de (C.L.H.N.); judith.schaffrath@uk-halle.de (J.S.); 4Krukenberg Cancer Center, University Hospital Halle, 06112 Halle, Germany; 5Institute of Pathology and Neuropathology, University of Tübingen, 72016 Tübingen, Germany; Achim.Rau@med.uni-tuebingen.de; 6Fraunhofer Institute for Cell Therapy and Immunology, 04103 Leipzig, Germany

**Keywords:** MDS, immune cell repertoire, prognosis, multiplex immunohistochemistry, CD34^+^ blasts

## Abstract

**Simple Summary:**

Despite a relationship between immune dysregulation and the course of myelodysplastic syndromes (MDS) has been discussed, a detailed understanding of this phenomenon is still missing. Therefore, multiplex analyses of bone marrow biopsies (BMB) from patients with MDS and secondary acute myeloid leukemia (sAML) were performed in order to determine the repertoire of lymphocyte subpopulations and their distance to CD34^+^ blasts. In MDS and sAML samples, the composition, quantity, and spatial proximity of immune cell subsets to CD34^+^ blasts were heterogeneous and correlated to the blast counts, but not to the genetics of the diseases, while in non-neoplastic BMB no CD8^+^ and FOXP3^+^ T cells and only single MUM1p^+^ B/plasma cells were detected in a distance of ≤10 μm to CD34^+^ hematopoietic progenitor cells (HPSC). We conclude that CD8^+^ and FOXP3^+^ T cells are not part of the immediate surrounding of CD34^+^ HPSC.

**Abstract:**

*Background*: Myelodysplastic syndromes (MDS) are caused by a stem cell failure and often include a dysfunction of the immune system. However, the relationship between spatial immune cell distribution within the bone marrow (BM), in relation to genetic features and the course of disease has not been analyzed in detail. *Methods*: Histotopography of immune cell subpopulations and their spatial distribution to CD34^+^ hematopoietic cells was determined by multispectral imaging (MSI) in 147 BM biopsies (BMB) from patients with MDS, secondary acute myeloid leukemia (sAML), and controls. *Results*: In MDS and sAML samples, a high inter-tumoral immune cell heterogeneity in spatial proximity to CD34^+^ blasts was found that was independent of genetic alterations, but correlated to blast counts. In controls, no CD8^+^ and FOXP3^+^ T cells and only single MUM1p^+^ B/plasma cells were detected in an area of ≤10 μm to CD34^+^ HSPC. *Conclusions*: CD8^+^ and FOXP3^+^ T cells are regularly seen in the 10 μm area around CD34^+^ blasts in MDS/sAML regardless of the course of the disease but lack in the surrounding of CD34^+^ HSPC in control samples. In addition, the frequencies of immune cell subsets in MDS and sAML BMB differ when compared to control BMB providing novel insights in immune deregulation.

## 1. Introduction

Myelodysplastic syndromes (MDS) are heterogeneous clonal hematologic diseases characterized by an ineffective hematopoiesis, one or more lineage dysplasia and peripheral cytopenia [1,2,3]. Multifactorial pathogenic features with diverse cytogenetic, molecular, and epigenetic alterations are associated with a variable clinical outcome [4,5]. Chronic inflammatory diseases associated with activated immune signaling pathways occasionally precede the clinical manifestation of MDS suggesting an etiopathogenetic link between chronic immune signaling, impaired stem cell quality, and alteration of the stem cell microenvironment [6,7]. There is growing evidence of immune deregulation in MDS pathogenesis [8,9], such as overexpression of immune-related genes in hematopoietic stem and progenitor cells (HSPC), abnormal activation of the innate immune pathways as well as aberrant immune responses [10,11]. However, the interrelationship between chronic immunologic stimulation and initiation as well as progression of MDS to secondary acute myeloid leukemia (sAML) remains largely unknown. Thus, a connection between immune deregulation [12,13,14] and established prognostic parameters, such as bone marrow (BM) blast count, cytogenetic alterations and the degree of peripheral cytopenia, which are all integrated in the Revised International Prognostic Scoring System (IPSS-R) [5,15], might contribute to a better understanding of the pathogenesis and consequences of altered immunologic features in this disease. Furthermore, this knowledge might improve the patient stratification, the use of current treatment options as well as the design of novel therapies.

There exists evidence that low risk MDS are related to higher levels of CD8^+^ cytotoxic T lymphocytes (CTL) and lower levels of FoxP3^+^ regulatory T cells (Treg), while the frequency of CTL and Treg is inversely correlated in high risk MDS [16,17,18,19,20]. Recently, a leukemia subtype specific immunological landscape was described suggesting a disease-specific immune regulation [21]. Next to T cell responses, deregulated B cell responses have an important impact on the pathogenesis of MDS [22,23,24] with a significantly higher somatic hypermutation rate of immunoglobulin heavy chain (IGH) clones in BM cells from del(5q) MDS patients indicating an extended number of antigen experienced B cells [25]. Thus, the inflammatory pattern mediated by the surrounding microenvironment might contribute to the mutagenic microenvironment in MDS [26].

To get deeper insights into the risk dependent deregulation of the lymphocyte subpopulations in MDS and sAML, a six-color multispectral imaging (MSI) panel was established to determine the frequency of immune cell subpopulations, their spatial distribution and the probability of the cellular interaction in 147 BM specimen from clinically and genetically characterized MDS and sAML patients as well as from control samples without evidence for hematological disease. Despite the use of next generation sequencing (NGS) technologies, for example, a systematic immunity screening approach of BM biopsies (BMB) of MDS patients analyzing the landscape of immune cells and CD34^+^ blasts is necessary to understand host cellular immune responses, which might be a key in MDS/sAML initiation and progression.

## 2. Results

### 2.1. Differences in the Frequency and Spatial Distribution of Immune Cell Subpopulations to CD34^+^ HSPC/Blasts in BMB from MDS, sAML, and Control Samples

In order to determine the composition and spatial distribution of lymphocyte subpopulations to CD34^+^ cells in BMB, the MSI technology was first adapted to decalcified FFPE BM samples. Using the established protocol [27], BMB from 45 controls, 69 MDS, and 33 sAML patients were analyzed by simultaneous staining for the markers CD3, CD8, FOXP3, MUM1p, and CD34, combined with nuclear staining using DAPI as representatively shown in Figure 1A. Patient characteristics are summarized in Table 1. One MSI field (1872 × 1404 pixel, 0.5 µm/pixel) contained a minimum of 2892 cells in the control BMB and up to 9297 cells in the sAML BMB. Within the BMB of control samples, the CD34^+^ HSPC counts varied between 0.7–1.8%, in BMB from MDS and sAML between 3.5–80.0%, respectively. Based on the staining pattern, four distinct immune cell subpopulations could be separated: CD3^+^CD8^+^ T cells, CD3^+^CD8^−^FOXP3^−^ T cells (T helper cells, and/or NKT cells), CD3^+^FOXP3^+^ T cells and MUM1p^+^CD3^−^ post-germinal center B/plasma cells. There was no statistically significant correlation between the frequency of leukocytes within the BM and the leukocyte count in the peripheral blood. However, slightly increased numbers of MUM1p^+^CD3^−^ post-germinal center B/plasma cells (*p* = 0.058) and CD3^+^ T cells (*p* = 0.061) were found in MDS patients with elevated peripheral blood leukocyte counts. Based on the evaluation of the distances between distinct immune cell subsets and between immune cells and CD34^+^ HSPC/blasts four distinct categories were defined as schematically demonstrated in Figure 1A. The spatial evaluation of the MSI data showed a heterogeneous histotopography of the immune cell infiltrate, in particular when comparing the control BMB and the MDS/sAML BMB, as representatively shown for two different specimens of control and MDS samples in Figure 1B–E.

### 2.2. Differences in the Frequency and Histotopography of Immune Cell Subsets in Spatial Relationship with CD34^+^ HSPC/Blasts in MDS, sAML, and Control Samples

The frequency and spatial distribution of immune cells in relation to CD34^+^ blast strongly varied in MDS vs. sAML. In control samples, no CD3^+^FOXP3^+^ T cells within a distance of <25 µm (*p* < 0.001) and no CD3^+^CD8^+^ T cells within a radius of ≤10 µm (*p* < 0.001) from CD34^+^ HSPC were detected (Figure 2). In addition, the frequency of MUM1p^+^CD3^−^ B/plasma cells and CD3^+^CD8^−^ T cells in the proximity of CD34^+^ HSPC was significantly lower (*p* < 0.001) for all categories in the control samples (Figure 2, for detailed data see Appendix A).

In contrast, BMB of MDS/sAML patients exhibited the different immune cell subpopulations analyzed in the proximity to CD34^+^ blasts (Figure 2). This effect was most pronounced for CD3^+^CD8^+^ T cells with an average of 0.33 CD3^+^CD8^+^ T cells (range 0–1.80) within a 10 µm radius around CD34^+^ blasts in sAML and an average of 0.08 CD3^+^CD8^+^ T cells (range 0–1.01) in MDS samples (*p* = 0.001). Furthermore, an average of 0.25 CD3^+^FOXP3^+^ T cells (range 0–1.53) was detected within a 10 µm radius around CD34^+^ blasts in sAML samples when compared to an average of 0.12 CD3^+^FOXP3^+^ T cells (range 0–1.10) in MDS BMB (*p* = 0.001). Detailed data are shown in Appendix A.

### 2.3. Correlation of the Frequency and Histotopography of Immune Cell Subsets to CD34^+^ Blasts in MDS and sAML

To exclude that differences in the frequency and distribution of the immune cell subpopulations analyzed were due to the overall BM cellularity, which was higher in BMB of MDS with excess of blasts (EB) and sAML, the cellularity of BMB obtained from controls and MDS patients without EB were compared. MDS samples without EB presented a comparable BM cellularity to control samples, but had significantly higher T cell counts in proximity to CD34^+^ blasts with 2 CD3^+^CD8^+^ T cells and 1.5 CD3^+^FOXP3^+^ T cells next to CD34^+^ blasts within the 10 µm radius. In contrast, a complete absence of CD3^+^FOXP3^+^ T cells and CD3^+^CD8^+^ T cells within a distance of <25 µm to CD34^+^ HSPC were found in control samples (Figure 2 and Figure 3; Appendix A).

BMB from controls and MDS/sAML patients exhibited only minor differences regarding the frequencies of respective immune cell subsets within the BM (Figure 3A), but demonstrated major differences in the spatial distribution of the immune cell subsets and the CD34^+^ HSPC/blasts, respectively (Figure 3B,C). CD3^+^FOXP3^+^ and CD3^+^CD8^−^ T cells were significantly more frequent in close spatial proximity to CD34^+^ blasts in sAML BMB when compared to that of MDS BMB. These data were most significant in sAML samples with CD34^+^ blast counts >30% with a significant increase (*p* = 0.022) in CD3^+^CD8^+^ T cells when compared to other disease subgroups and controls. Furthermore, a significant higher frequency of MUM1p^+^CD3^−^B/plasma cells was found in sAML cases with a blast count of >30%. However, within the MDS subgroups of different blast counts, the frequency of these cells in the neighborhood of CD34^+^ blasts was nearly stable suggesting no direct linear correlation between blast counts and quantity of immune cell subpopulations in the proximity to CD34^+^ blasts.

Samples analyzed were separated into following groups: control BMB (controls), MDS without excess of blasts (MDS < 5%) encompassing MDS-SLD, MDS-RS-SLD, MDS-MLD, MDS-RS-MLD; MDS with excess of blasts 1 (MDS-EB1); MDS with excess of blasts 2 (MDS-EB2); sAML with <30% blasts (sAML < 30% blasts) and with >30% blasts (sAML > 30% blasts). 

### 2.4. Effect of the Mutational Status in MDS on the Frequency and Histotopography of the Immune Cell Subsets

In order to investigate whether the landscape of genetic mutations of MDS affects the immune cell composition of the BM and the outgrowth control of malignant cells, targeted NGS was carried out on the samples from MDS (n = 28) and sAML (n = 10) patients. As summarized in Table 2, mutations were detected in 10/22 genes analyzed. These included in particular genes involved in the epigenetic regulation, signal transduction, transcription, and DNA repair with the highest frequency of mutations in TP53 (10/28), which might contribute to genomic instability and consequently to a selective immune pressure [28]. Mutations were detected in 50% of MDS patients and varied from five mutations in one to no mutation in three out of 28 patients (Table 2).

Next to samples with mutations in genes coding for proteins involved in the signal transduction, increased frequencies of CD3^+^CD8^+^ T cells (*p* = 0.058) and CD3^+^ FOXP3^+^ T cells (*p* = 0.021) and a slight, but not significantly increased proportion of CD3^+^CD8^−^ T cells in patients with mutations in splicing factors (*p* = 0.053) and TP53 (*p* = 0.102) were found, while neither significant differences in the frequency nor in the spatial distribution of the immune cell subpopulations in relation to CD34^+^ blasts were detected. In contrast, in patients harboring one or more mutations in splicing factors, chromatin modification, and DNA methylation, a higher frequency of CD3^+^CD8^−^ T cells and CD3^+^FOXP3^+^ T cells were detected in close proximity to CD34^+^ blasts in all distance categories (*p* = 0.017). No significant association between the mutational status including TP53 mutations and the progression to sAML was detected.

### 2.5. Influence of Cytogenetic Alterations and the Frequency and Histotopography of the Immune Cell Subsets in MDS

Based on the cytogenetic aberrations in MDS without EB and EB1-2, patients were classified according to the CCSS [15] (Table 1). The lower risk subgroup includes samples with very low and low risk cytogenetic aberrations (CCSS 1–2), while diseases with intermediate, high, and very high cytogenetic scores were defined as higher risk subgroup (CCSS 3–5).

With respect to the different immune cell subpopulations, the overall frequency of CD3^+^FOXP3^+^ T cells and MUM1p^+^CD3^−^ B/plasma in relation to the total of number of cells in the BMB was significantly higher in CCSS high risk cases compared to CCSS low risk cases (*p* = 0.042; *p* = 0.004; Figure 4A). However, neither significant differences in the overall frequency of CD3^+^CD8^+^ nor CD3^+^CD8^−^ T cells were detected between CCSS low risk, CCSS high risk and control cases (Figure 4A).

Regarding the spatial distribution of the immune cell subpopulations in relation to CD34^+^ blasts no significant differences in the direct and close contact to CD34^+^ blasts (<10 μm) was found when BMB with CCSS 1–2 and CCSS 3–5 were compared (Figure 4B; Appendix A). However, within a 25 µm radius to the CD34^+^ blasts significant more CD3^+^FOXP3^+^ T cells (*p* = 0.041) and MUM1p^+^CD3^−^ B/plasma cells (*p* = 0.012) and increased numbers of CD3^+^CD8^+^ T cells (*p* = 0.099) were detected in the higher risk subgroup when compared to all other categories (data shown in Appendix A).

### 2.6. Spatial Distribution of Immune Cell Subsets and CD34^+^ Blasts in Relation to MDS Progression to sAML

A possible prognostic value of the immune cell landscape in association with CD34^+^ cells was postulated. Indeed, patients with a higher IPSS-R (intermediate, high and very high risk) showed increased frequencies of CD3^+^FOXP3^+^ T cells (*p* = 0.004) and MUM1p^+^CD3^−^ B/plasma cells (*p* = 0.062) in relation to the total of cells in the BM (Figure 4C). Concerning their spatial distribution, increased numbers of CD3^+^FOXP3^+^ T cells within a 10 µm radius of CD34^+^ blasts (*p* = 0.016), and CD3^+^FOXP3^+^ T cells within a 10 µm radius of CD3^+^CD8^+^ T cells (*p* = 0.04) were found in IPSS-R higher risk groups (Figure 4D, Appendix A). To exclude therapy effects, BMB of patients with prior HMA treatment and BMB of patients treated with ASCT in the course of the disease were excluded as well. The remaining patients were separated into patients with (n = 6) and without (n = 29) progress to sAML in the course of MDS (Table 1). While the mean frequency of T cell subsets was comparable in both subgroups, the frequency of MUM1p^+^CD3^−^ B/plasma cells in relation to the total of cell number in the BM was significantly higher in patients with disease progression (*p* = 0.033). However, no significant differences in the spatial distribution of the immune cells in relation to the CD34^+^ blasts as well as to the respective immune cell subsets were found. Interestingly, patients without progress in the period of four years after diagnosis showed more CD3^+^FOXP3^+^ T cells in close spatial proximity of CD3 ^+^ CD8^+^ T cells (*p* = 0.053, see Appendix A).

## 3. Discussion

Due to the complexity of MDS, several genetic, immunologic as well as environmental factors have been shown to play a role in the pathophysiology of this disease, recent evidence suggest that immune dysregulation is a central feature of MDS and linked to disease initiation and progression to sAML [29,30]. So far, most studies analyzed the influence of the innate immune response [16], in particular the role of impaired Toll-like receptor (TLR) signaling pathways due to overexpression and/or mutations in immune related genes and microRNAs [30,31]. Furthermore, a high frequency of mutations in DNMT3A, TET2, ASXL1, and splicing factors, frequently identified in MDS and sAML, can influence the innate immune signaling and promote NF-κB signaling through various mechanisms [32,33,34]. In contrast, only limited information concerning the influence of the adaptive immune system, in particular T cell surveillance, in MDS biology exists [16]. In this context, it was demonstrated that in low-risk MDS a decreased number of Treg was associated with effector T cell targeting of HSPC, while in high-risk MDS an increased frequency of Treg led to an immune suppressive microenvironment [35]. Furthermore, there exist data demonstrating that changes in the number and functionality of T cells correlated with the release of inflammatory cytokines, but also with other soluble constituents, like chemokines and hormones [20,36], leading directly or indirectly to the suppression of CD4^+^ and CD8^+^ T cells [37].

In addition, it has been hypothesized that the interaction of MDS blasts and the BM microenvironment might play an important role for disease homeostasis. More precisely, a bidirectional crosstalk between MDS clone and BM microenvironment seems to maintain the malignant clone, but also shape the BM environment [38].

During the last decade, multiparameter flow cytometry as well as high throughput technologies, like RNAseq or Nanostring analysis, have been applied for the characterization of the immune landscape of MDS and AML in peripheral blood and BM [39]. This technologies were further employed for the monitoring of immune modifying agents in high risk MDS/AML [40]. Immunophenotyping demonstrated a dynamic immune cell repertoire with a higher frequency of NK cells and CTL in low risk MDS patients when compared to controls, while an increased Treg frequency was found in high risk MDS patients [18,20]. With the availability of CyTOF, immunophenotyping of immune cell subsets and HSPCs could be extended. However, interactions of adaptive immune cells within the BM might affect the initiation and progression of MDS. Therefore, in depth analyses of the local frequency, histotopography, and the spatial interaction of adaptive immune cell subpopulations with hematopoietic cells were performed in this study using MSI. Although the number of cells characterized in BMB using MSI is lower when compared to flow cytometry, this technique allows to visualize surface membrane, cytoplasmic, and nuclear protein expression and quantifies the spatial relationships between cells. The latter was assessed by the characterization of four distance categories.

Despite this limitation, the tissue architecture was analyzed using antibodies directed against CD3, CD8, FOXP3, and MUM1p to label immune cell subsets and CD34 to label HSPC and MDS/sAML blasts. With the exception of a recent study there exists no general information, whether spatial distances between immune cell subsets and in relation to tumor cells are of biological relevance in hematologic malignancies [41]. In solid tumors, a prognostic impact of tumor and immune cell proximity was shown within an intercellular distance of 20–30 µm [42,43]. However, it is noteworthy that the immunoscore based on the quantification of cytotoxic and memory T cells in the tumor center and tumor margin was generated in solid tumors, but cannot be directly applied to BM samples of MDS/sAML due to the lack of an invasive margin and tumor core in these specimens. In this study, most significant differences in the samples analyzed regarding immune cell subsets and blasts were found within a distance of <10 µm suggesting a biological relevance.

With focus on the absolute frequency of the different immune cell subpopulations in relation to the entire BM cells, the comparison of control BMB and MDS/sAML BMB did neither reveal significant differences concerning the number of CD3^+^ T cells nor the number of MUM1p^+^ B -/plasma cells. However, MSI demonstrated a complete absence of CD8^+^ and FOXP3^+^ T cells within a radius of 10 µm and a highly significant lower frequency of MUM1p^+^CD3^−^ B/plasma cells within a radius of <50 µm to CD34^+^ HSPC in control samples, while in MDS/sAML BMB, these cells were regularly detected in these localizations. These findings could neither be explained by a lower BM cellularity in control samples nor by a falsely consideration of CD34^+^ endothelial cells, since this cell type was excluded by the typical luminal CD34 positivity of the endothelial cells. Therefore, these data point again to the advantage of MSI by focusing on the histotopography between different cell types.

The complete absence of CD3^+^CD8^+^ and CD3^+^FOXP3^+^ T cell subsets in the neighborhood of CD34^+^ hematopoietic cells postulates that these immune cell subpopulations are not common components of the normal HSC niche mainly consisting of mesenchymal cells [44,45,46,47,48,49,50]. Vice versa, the presence of these immune cell subsets within the CD34^+^ stem cell niche in the BM of MDS and sAML patients indicates an impact of these cells in the pathogenesis of both malignancies. This is in accordance with the accumulating evidence that the BM microenvironment is a regulator of the neoplastic stem cell pool and key mediator of MDS pathophysiology [45,47,48,50]. A direct involvement of the T cell subsets in relation to CD34^+^ blasts and their role in the biology of MDS/sAML is further supported by the finding that these cells were detected with the highest frequency in sAML patients with a blast count of >30%, a threshold known to describe a more accelerated subset of sAML. Based on xenograft models, MDS cells require support from microenvironmental components to propagate disease. Furthermore, MDS-derived MSC are molecularly distinct from their healthy counterparts in terms of their gene expression profiles [49], in particular regarding inflammation-associated genes and expression of immune modulatory molecules, like galectin implicated in inhibition of adaptive immune system [26,51].

Cytogenetic aberrations and frequently occurring mutations with known prognostic value in MDS might lead to the generation of neoantigens with the consecutive generation of antigen-specific immune responses in MDS and sAML [52]. Significant higher levels of CD3^+^FOXP3^+^ T cells in patients with higher risk cytogenetic aberrations (CCSS score 3–5), which are associated with the presence of myeloid derived suppressor cells are also known to contribute significantly to the dysregulation of immune surveillance in MDS [53]. In accordance to the presented data here, other studies showed that patients with mutations in TP53 and genes involved in the signal transduction presented slightly increased numbers of T cells within the BM. which has in some cases a prognostic value in patients with proved TP53 mutations [54,55]. In addition, another study presented data demonstrating that patients with TP53 mutations exhibited a reduced number of helper T cells, but a significant increase in Treg and MDSC. These results suggest that the tumor microenvironment (TME) of TP53 mutated MDS and sAML might display an immune escape phenotype [56].

Most interestingly, in patients harboring more than one mutation in splicing factors, chromatin modifiers and DNA methylating features a significantly closer proximity of CD3^+^CD8^−^ T cells and CD3^+^FOXP3^+^ T cells to CD34^+^ blasts was seen. Since most of the mutations identified alter the adaptive rather than the innate immune signaling, we conclude that the presence of T and B/plasma cell subpopulations in close proximity to CD34^+^ blasts in MDS and sAML is independent of the mutation induced neoantigens associated with immune modulatory mechanisms [34,46].

In conclusion, immune profiling of HSC niches and spatial immune cell interactions by MSI represents a powerful tool for investigating the clinical relevance of the frequency and the spatial distribution of immune cell subpopulations, showing significant differences in the spatial histotopography of immune cell subsets in BMB of healthy donors, MDS, and sAML. However, beyond the biological relevance shown by the spatial relationship in this study, further investigations to identify the underlying immune regulatory mechanisms and the functional relevance of spatial proximity of these immune cell and hematopoietic cell subsets for the design of potent immunotherapies in MDS and sAML are urgently needed.

## 4. Materials and Methods

### 4.1. Patient Characteristics

Bone marrow biopsies (BMB) collected between 2014 and 2019 at the Medical Faculty of the Martin-Luther University Halle-Wittenberg, Germany, were part of the routine diagnostic approach based on screening and/or treating patients within clinical trials. The cohort consists of BMB from 102 patients divided in 69 MDS patients with different blast counts (without and with excess of blasts) and 33 BMB of sAML patients. All MDS/sAML specimen presented myeloid blasts with a CD34^+^ immunophenotype and their clinico-pathological characteristics are summarized in Table 1. Furthermore, 45 BMB from age-matched patients with normal blood cell count and without evidence of myeloid or lymphoid neoplasia within the BMB served as control samples. Clinical, laboratory, molecular and cytogenetic data including the CCSS, risk group according the IPSS-R and treatment data were collected from the medical records. MDS patients were stratified into the following disease groups: MDS without EB (<5% blasts), encompassing MDS with single lineage dysplasia (MDS-SLD), MDS with ring sideroblasts and single lineage dysplasia (MDS-RS-SLD), MDS with multi-lineage dysplasia (MDS-MLD), MDS with ring sideroblasts and multi-lineage dysplasia (MDS-RS-MLD); MDS with EB-1 (MDS-EB-1, 5–9.9% blasts); MDS with EB-2 (MDS-EB-2, 10–19.9% blasts) and sAML (≥20% blasts) based on the cytological blast count. sAML patients were further divided in two groups characterized by BM blast counts of 20–29.9% and higher than 30%.

### 4.2. Study Approval

Informed consent was obtained from all patients for the use of their diagnostic material for scientific research and the use of FFPE tissue samples from patients was approved by the Ethical Committee of the Medical Faculty of the Martin Luther University Halle-Wittenberg, Halle, Germany (2017-81).

### 4.3. Standard Morphological Evaluation of the Bone Marrow

Diagnosis of MDS and sAML with myelodysplasia related changes was performed according to the diagnostic criteria of the World Health Organization (WHO) classification of Tumors of Hematopoietic and Lymphoid tissues, fourth edition 2017 [57]. To confirm the diagnosis of MDS and sAML, conventional histological and cytological examination as well as immunohistochemistry (IHC) was performed. Monoclonal antibodies (mAb) directed against CD34 (clone QBend/10, Thermo Fisher Scientific, Fremont, CA, USA), CD117 (clone CD117, c-kit A4502, Dako, Santa Clara, CA, USA), MPO (clone myeloperoxidase A0398, Dako, Santa Clara, CA, USA), lysozyme (EP134, Epitomics, Burlingame, CA, USA), and CD71 (MRQ-48, Cell Marque, Rocklin, CA, USA) were used according to the supplier’s instructions.

### 4.4. Multispectral Imaging (MSI)

The frequency, localization and spatial proximity of immune cell subpopulations, CD34^+^ hematopoietic stem and progenitor cell (HSPC) and of CD34^+^ blasts in case of malignancy were analyzed by MSI. The staining procedure was performed as recently described [27,58]. The marker panel used for staining included mAb directed against CD34 (QBend, Labvision, Germany, 1:500, pH6), CD3 (Labvision, Germany, clone SP7), CD8 (Abcam, Cambridge, UK, clone SP16), FOXP3 (Abcam, UK, clone 236A/E7), and MUM1p (Dako, USA, cloneMUM1p). Briefly, all primary mAb were incubated for 30 min. Tyramide signal amplification (TSA) visualization was performed using the Opal seven-color IHC kit containing fluorophores Opal 540, Opal 570, Opal 620, Opal 650, Opal 690 (Perkin Elmer Inc., Waltham, MA, USA), and DAPI. Stained slides were imaged employing the PerkinElmer Vectra Polaris platform. To unify the spatial distribution analysis three ×20 MSI fields (1872 × 1404 pixel, 0.5 µm/pixel) were manually selected on each slide based on representativeness and tissue size. Since the BMBs showed a high variability in quality and size, areas with preserved architecture were chosen, while hemorrhagic areas and areas with artificial lacks were excluded. Cell segmentation and phenotyping of the cell subpopulations were performed using the inForm software (PerkinElmer Inc., USA). The frequency of all immune cell populations analyzed and the cartographic coordinates of each stained cell type were obtained. The spatial distribution of cell populations was analyzed using PerkinElmer inform and R script for immune cell enumeration and relationship analysis. The multiplex staining panel could differentiate the distinct T cell subpopulations into CD3^+^CD8^+^ T cells, CD3^+^CD8^−^ T cells and CD3^+^FOXP3^+^ T cells, respectively, without providing information about T cell function, such as e.g., cytotoxicity. All MUM1p^+^ B cells/plasma cells were CD3 negative. CD34^+^ blast cells were separated from CD34^+^ endothelial cells by histomorphology based on their localization and cytological appearance. The inform software was trained to exclude endothelial cells with incomplete line-shaped luminal CD34 expression and include CD34^+^ blasts with membranous CD34 expression.

### 4.5. Mutational Analysis—Targeted Next Generation Sequencing

Targeted mutation analyses were performed by Next Generation Sequencing (NGS; Ion GeneStudio S5 prime, Thermo Fisher Scientific, Waltham, MA, USA) using an AmpliSeq custom panel designed for myeloid disorders comprising hotspot regions in 21 genes (*JAK2*, *FLT3*, *STAT3*, *ASXL1*, *IDH1*, *IDH2*, *SRSF2*, *SF3B1*, *U2AF1*, *SETBP1*, *MPL*, *KIT*, *CBL*, *CSF3R*, *CALR*, *ETNK1*, *KRAS*, *NRAS*, *HRAS*, *BRAF*, *GNAS*) and the 10 genes (*CEBPA*, *RUNX1*, *IKZF1*, *DNMT3A*, *EZH2*, *ZRSR2*, *TP53*, *TET2*, *NPM1*, *STAG2*). Amplicon library preparation and semiconductor sequencing was done according to the manufacturers’ manuals using the Ion AmpliSeq Library Kit v2.0, the Ion Library TaqMan Quantitation Kit, the Ion 510 & Ion 520 & Ion 530 Kit—Chef and the Ion 520 Chip Kit (Thermo Fisher Scientific, Fermont, CA, USA). Variant calling of non-synonymous somatic variants compared to the human reference sequence was performed using Ion Reporter Software (Thermo Fisher Scientific, Version 5.12.3.0). Variants were filtered with a threshold allele frequency of 5%.

Variants called by the Ion Reporter Software were visualized using the Integrative Genomics Viewer (IGV; Broad Institute, Cambridge, MA, USA; Version 2.5.2) to exclude panel-specific artefacts.

### 4.6. Statistics

Statistical analyses were performed employing IBM SPSS Statistics. Kolmogorov–Smirnov test revealed non-parametric data (*p* < 0.05). The Mann–Whitney U test was employed to compare clinical data, frequencies of immune cell subpopulations and their spatial distribution. *p* values < 0.05 were considered statistically significant. The figures were generated using the GraphPad Prism 7.0 software.

## 5. Conclusions

In conclusion, immune profiling of HSC niches and spatial immune cell interactions by MSI represents a powerful tool for investigating the clinical relevance of the frequency and the spatial distribution of immune cell subpopulations in BMB of healthy donors, MDS and sAML. However, further investigations to identify the underlying immune regulatory mechanisms and their importance for the design of potent immunotherapies in MDS and sAML are urgently needed.

## Figures and Tables

**Figure 1 cancers-13-00186-f001:**
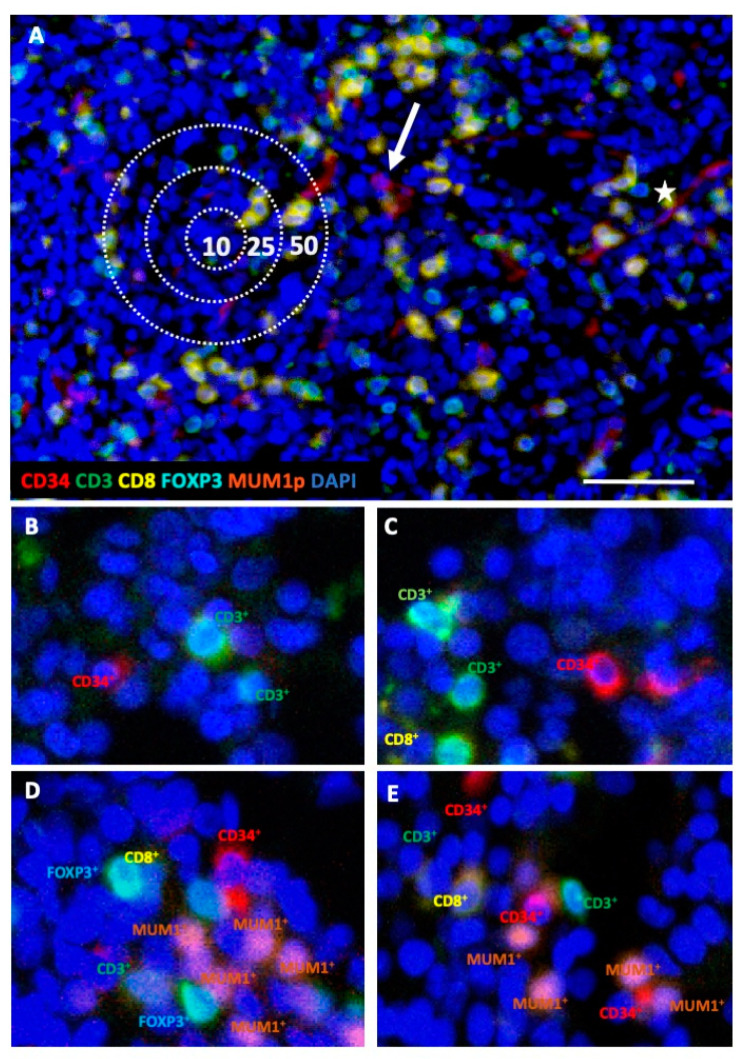
Representative multiplex immunohistochemistry (IHC) of BMB from controls and MDS patients. All BMB were stained with a five-color antibody panel consisting of CD34 (red), CD3 (green), CD8 (yellow), FOXP3 (turquoise), MUM1p (orange) antibodies, and counterstained with DAPI (blue). (**A**) Representative BMB with MDS with multi-lineage dysplasia without excess of blasts (MDS-MLD). The intercellular distance algorithm with four categories was defined as follows: (i) direct intercellular contact, (ii) cells within a radius of 10 µm, (iii) of 25 µm, and (iv) of 50 µm; the scale bar is 50 µm. One representative CD34^+^ blast with circular membranous staining pattern is highlighted with an arrow. In comparison, CD34^+^ endothelial cells with luminal line-shaped positivity are marked with a star. (**B**,**C**) two representative control BMB with a homogeneous staining pattern comprising only few CD3^+^CD8^−^ T cells, but no CD3^+^CD8^+^ and no CD3^+^FOXP3^+^ T cells in a radius of <10 μm to CD34^+^ HSPC. (**D**,**E**) two representative MDS samples are presented, which displays variable, but generally higher frequency of different T and B/plasma cell subpopulations in close proximity of the CD34^+^ blasts.

**Figure 2 cancers-13-00186-f002:**
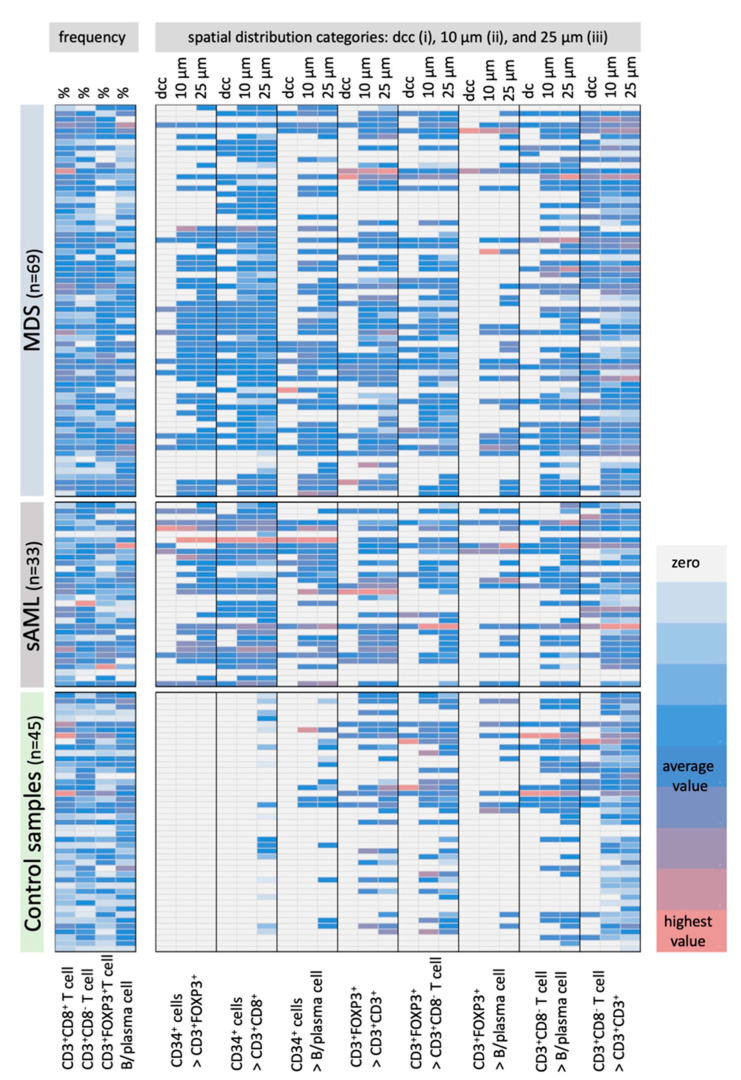
Frequency and the spatial distribution of immune cells analyzed in BMB from controls and MDS/sAML patients in correlation to CD34^+^ blasts. MSI data of 45 control samples, 65 MDS and 33 AML BMB were evaluated regarding the frequencies of CD3+CD8+ T cells, CD3+CD8− T cells, CD3+FOXP3+ T cells and MUM1p+CD3− B/plasma cells as well as their spatial distribution to each other and to the CD34+ HSPC/blasts (CD34+ blasts in relation (>) to CD3+FOXP3+ T cells, CD3+CD8+ T cells, MUM1p+CD3− B/plasma cells; CD3+FOXP3+ T cells in relation (>) to CD3+CD8+ T cells, to CD3+CD8− T cells MUM1p+CD3− B/plasma cells, and CD3+CD8− T cells in relation (>) to MUM1p+CD3− B/plasma cells and CD3+CD8− T cells) and results are represented in a heat map. All spatial relations are depicted according to the categorization in direct cellular contact (dcc), the 10 µm and the 25 µm radius, respectively. The number of cells in each spatial category or the frequency of the immune cells are color-coded, in which white/light grey codes for lowest value, blue codes for average value and light red codes for highest value of immune cell subsets. Most remarkable is the presence of only few B/plasma cells and no CD3+CD8+ and CD3+FOXP3+ T cell subpopulations in the proximity to CD34+ HSPC in the control samples, while the frequency of these immune cell subpopulations is comparable within the BM of MDS and sAML samples.

**Figure 3 cancers-13-00186-f003:**
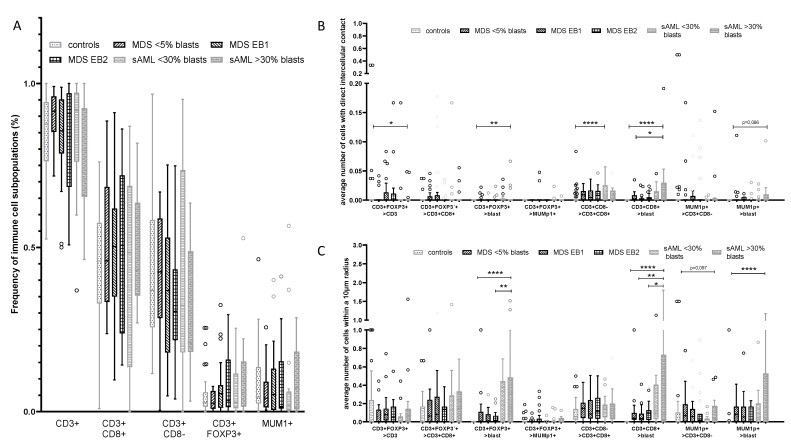
Overall frequency and composition of the immune cell infiltrate with respect to the CD34^+^ blast cell count in MDS and sAML. (**A**) Determination of the frequency of CD3+ T cells, CD3+CD8+ T cells, CD3+CD8− T cells, CD3+FOXP3+ T cells and MUM1p+CD3− B/plasma cells in relation to the total of cells within the BMB demonstrating minor differences within the respective groups. In contrast, the spatial distribution of immune cell subsets in relation to the CD34+ blast frequency in BMBs of MDS and sAML patients compared to controls revealed a significantly elevated frequency of MUM1p+CD3− B/plasma cells in sAML cases with >30% CD34+ blasts when compared to all other diseased patients. Analysis of the different intercellular distances (categories I–IV) demonstrated the most significant differences within a radius of 10 µm [II]. Representative data are shown for the parameters direct intercellular contact (**B**) and intercellular distance of 10 µm (**C**). Statistically significant relations are marked with asterisks (* *p* < 0.05; ** *p* <0.005, *** *p* < 0.0005).

**Figure 4 cancers-13-00186-f004:**
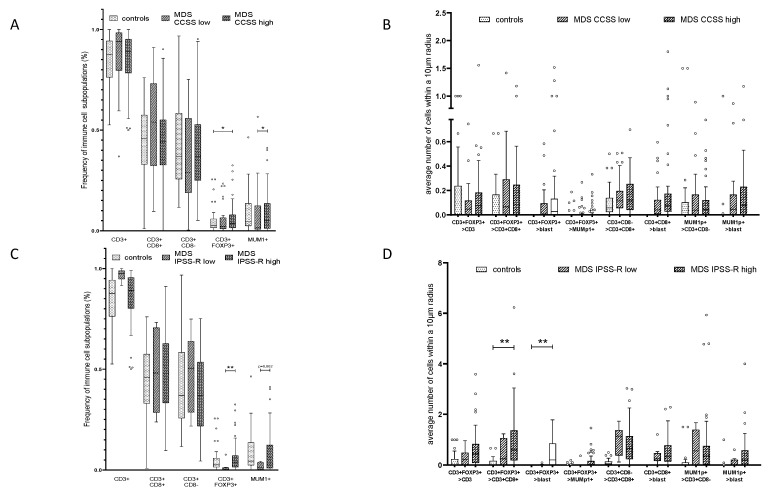
Composition and frequency of the immune cell infiltrate and their spatial distribution to CD34^+^ blasts in MDS/sAML in association to the karyotype and the IPSS-R. (**A**) Correlation of MDS with CCSS low and high to the frequency of MUM1p+CD3− post–germinal center B cells/plasma cells, CD3+CD8+, CD3+CD8−, and CD3+FOXP3+ T cells, presented as number of immune cell subpopulations in % with a significant correlation of MUM1p+CD3− post–germinal center B cells/plasma cell frequency/CD3+FOXP3+ T cells and CCSS high. (**B**) Correlation of CCSS high versus CCSS low MDS specimen considering the average number of immune cell subpopulations within a radius of 10 µm around CD34+ blasts revealed no significant differences between the CCSS high versus CCSS low prognostic group. (**C**) Correlation of MDS with IPSS-R risk score low and high to the frequency of MUM1p+CD3− post-germinal center B cells/plasma cells, CD3+CD8+, CD3+CD8−, and CD3+FOXP3+ T cells, presented as number of immune cell subpopulations in % with a significant correlation of CD3+FOXP3+ T cells and slightly increased numbers of MUM1p+CD3− B/plasma cells in IPSS-R high risk cases. (**D**) Spatial analysis data for IPSS-R high versus IPSS-R low in untreated MDS specimen considering average number of immune cell subpopulations within a radius of 10 µm revealed no significant differences. Statistically significant relations are marked with asterisks (* *p* < 0.05; ** *p* <0.005, *** *p* < 0.0005).

**Table 1 cancers-13-00186-t001:** Clinico-pathological characteristics and sample specifications in terms of risk factors, treatment, and prognosis.

Category		MDS	sAML	Controls
Number of samples		69	33	45
Age		68.2 (42–86)	68.9 (47–82)	59.6 (39–84)
Sex	male	39	18	24
	female	30	15	21
Leukocyte count in peripheral blood (×10^9^/L)		4.5 (0.8–46.2)	5.6 (0.1–30.0)	8.1 (3.8–9.7)
Patients with (auto)immune disorders		10	2	0
	Autoimmune thyroiditis (Hashimoto thyroiditis, Graves’ disease)	5	1	-
	autoimmune cholangitis	1	-	-
	rheumatoid arthritis	3	-	-
	Myasthenia gravis	1	1	-
MDS	MDS-SLD, MDS-RS-SLD, MDS-MLD, MDS-RS-MLD	22	-	-
	MDS-EB1	26	-	-
	MDS-EB2	21	-	-
CCSS	1—loss of Y chromosome, del(11q)	7	2	-
	2—normal karyotype, del(5q), del(12p), del(20q)	27	5	-
	3—del(7q), gain of chromosome 8 and 19, isochromosome 17q	17	10	-
	4—gain of chromosome 3 and 7, complex (>2)	1	7	-
	5—complex (>3)	17	9	-
IPSS-R	1—very low risk	2	-	-
	2—low risk	6	-	-
	3—intermediate risk	20	-	-
	4—high risk	26	-	-
	5—very high risk	15	-	-
Treatment	patients without specified treatment prior to BMB extraction	35 of 69	-	-
	HMA treatment prior to BMB extraction	17 of 69	6 of 33	-
	ASCT treatment following BMB extraction	23 of 69	9 of 33	-
	MDS patients with progress to sAML in the course of the disease (regardless treatment)	18 of 69	-	-
	untreated MDS patients with progress to sAML in the course of the disease (without prior HMA treatment and without ASCT in the course of the disease)	6 of 35	-	-

MDS patients were categorized into different MDS subtypes as follows: MDS without excess of blasts (encompassing MDS with single lineage dysplasia (MDS-SLD), MDS with ring sideroblasts and single lineage dysplasia (MDS-RS-SLD), MDS with multi-lineage dysplasia (MDS-MLD), MDS with ring sideroblasts and multi-lineage dysplasia (MDS-RS-MLD)), MDS with excess of blasts 1 (MDS-EB1), and MDS with excess of blasts 2 (MDS-EB2). Cytogenetic aberrations were classified according to the Comprehensive Cytogenetic Scoring System (CCSS), while risk factors were classified according to the Revised International Prognostic Scoring System (IPSS-R). Treatment with hypomethylating agents (HMA) prior to BMB extraction and allogeneic stem cell transplantation (ASCT) treatment following BMB extraction are separately marked.

**Table 2 cancers-13-00186-t002:** Landscape of mutations and their prognostic relevance in MDS and progression to sAML.

**Patient Numbers**			1	2	3	4	5	6	7	8	9	10	11	12	13	14	15	16	17	18	19	20	21	22	23	24	25	26	27	28
**function**	**gene**	**(n/%)**																												
**cohesin**	*STAG2*	*0 / 0.0*																												
**DNA repair**	*TP*53	*10 / 35.7*			*	*						*	*	*	*							*		*			*			*
**chromatin**	*ASXL*1	*3 / 10.7*																*	*	*										
**modification**	*EZH*2	*2 / 7.1*													*														*	
**DNA**	*DNMT*3*A*	*2 / 7.1*																				*	*							
**methylation**	*IDH*1	*1 / 3.6*							*																					
	*IDH*2	*2 / 7.1*														*				*										
	*TET*2	*3 / 10.7*						*									*			*										
**RNA splicing**	*SRSF*2	*2 / 7.1*							*									*												
	SF3B1	*1 / 3.6*					*																							
	U2AF1	*0 / 0.0*																												
**signal**	ZRSR2	*0 / 0.0*																												
**transduction**	*BRAF*	*1 / 3.6*																									*			
	*CALR*	*0 / 0.0*																												
	*CBL*	*2 / 7.1*																*											*	
	*ETKN1*	*0 / 0.0*																												
	*CSFR3*	*0 / 0.0*																												
	*FLT3*	*1 / 3.6*																									*			
	*GNAS*	*0 / 0.0*																												
	*HRAS*	*0 / 0.0*																												
	*JAK2*	*2 / 7.1*									*										*									
	*KIT*	*0 / 0.0*																												
	*KRAS*	*0 / 0.0*																												
	*MPL*	*0 / 0.0*																												
	*NRAS*	*0 / 0.0*																												
	*STAT3*	*0 / 0.0*																												
**transcription**	*CEBPA*	*1 / 3.6*		*																										
**factors**	*IKZF1*	*0 / 0.0*																												
	*NPM1*	*1 / 3.6*		*																										
	*SETBP*1	*3 / 10.7*													*					*									*	
	*RUNX*1	*1 / 5.0*																												
**total number of mutations/patient**			0	2	1	1	1	1	2	0	1	1	1	1	3	1	1	3	1	4	1	2	1	1	0	0	3	0	3	1
**CCSS**			2	2	5	2	2	3	3	2	2	5	5	5	5	2	3	3	1	3	2	3	3	5	5	3	5	2	3	3
**blast count (%)**			4	3	7	11	4	7	4	3	3	14	8	3	7	8	9	7	4	4	4	4	9	18	17	8	11	19	14	16
**IPSS-R group**			2	3	4	2	3	4	4	3	1	5	5	5	5	2	4	4	3	4	3	4	5	5	5	3	5	5	4	5
**progress to sAML**			no	no	no	no	no	no	no	no	no	no	no	no	no	no	no	no	no	no	no	yes	yes	yes	yes	yes	yes	yes	yes	Yes

Twenty-eight samples from MDS patients were analyzed by targeted NGS as described in Materials and Methods. The detected mutations are marked with asterisks (*), their frequency shown as numbers and percentage and correlated to the cytogenetic aberrations using the Comprehensive Cytogenetic Scoring System (CCSS), the blast count (%) and the Revised International Prognostic Scoring System (R-IPSS) risk groups (1—very low risk to 5—very high risk).

## Data Availability

The data presented in this study are available on request from the corresponding author. The data are not publicly available due to ethical reasons.

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
