# Peer review of "Altered Spatial Composition of the Immune Cell Repertoire in Association to CD34+ Blasts in Myelodysplastic Syndromes and Secondary Acute Myeloid Leukemia"

_cancers, 2021, doi:10.3390/cancers13020186_

Round 1
Reviewer 1 Report
In this manuscript the Authors evaluated the spacial composition of the immune cell repertoire in the bone marrow of patients with MDS and secondary AML.
They found that the levels of CD3+CD8+/- FOXP3+/- T cells, as well as MUM1p+CD3- B cells were similar among patients' categories and as compared with healthy donors. However, they highlighted a different spatial proximity of immune cells in relation to CD34+ cells: significantly closer in patients than in controls, by multispectral imaging.
Although the topic is interesting, given the emergent role of the immune system in MDS/AML, the message is not clearly exposed, and the methods may be questioned.
1) The main result is the proximity of the immune cells to the CD34+ cells in marrow samples of patients as compared to controls. In the materials and methods section, they state that “CD34+ blast cells were separated from CD34+ endothelial cells by histomorphology based on their localization and cytological appearance”. This appears not enough to exclude that the vicinity of the immune cells to CD34+ ones by multispectral imaging relies on endothelial proximity (rather than on blast cells). If we assume that the “histomorphologic and cytologic” criterion is enough, the same could be stated for the identification of the immune cells and the strength of the study design would be further weakened.
2) The study does not answer, not even discuss, the question whether the immune cells establish a functional relationship/signaling/communication with the MDS stem cells: were electronic microscopy experiments performed? (even in the literature), are microparticles or other mediators implied? Some functional studies would be more necessary to clarify this issue.
3) the Authors may consider enriching the number of markers used to better identify the various immune cell populations (i.e. CD4, CD16, CD56, CD25, CD19, CD38 etc...) since all of them have different functional significances. If an extended evaluation is not performed, the functional role of immune cells close to CD34+ HSCs (if not vascular cells) remains unclear.
4) Table 1 contains scanty clinical information and may be implemented by adding hematologic values, response to therapy, positivity of autoimmunity markers, and death. Moreover, abbreviations should be better defined.
5) The numbers of “immune” cells in treated and untreated patients are given as “average” with no range, and is often < to 0. This Referee feels that it would be better to give values ad median (range) and to define the number (%) of patients/samples that displayed higher frequency of “immune cell populations”.
6) Overall, the results are too long and not clearly exposed. I would suggest to improve figures (i.e. 3B and 3C, and 4B and 4C) that are sometimes hardly readable, and to include a table where the reader can easily distinguish median number of each cell population for each distance considered (rows) for – all patients – treated – untreated – IPSS categories etc (columns).
7) The mutational analysis is interesting, but it is not the focus of the article and table 2 is nice but “out of the blue”. Data are discussed altogether (MDS and AML) and at the end of the paragraph the reader is confused: is there a relationship with presence and spatial distribution of “immune cells”? The association of response to therapy and somatic mutations appears out of the focus of the study and might be removed if not linked to the immune cells infiltrate.
8) The discussion appears as a small dissertation of previous evidence, without clearly showing the added value of the present study. An effort should made to build the discussion on Authors’ findings, addressing the strengths and weaknesses of the study, the future implications, and the likely pathogenic, prognostic and therapeutic impact
Page 8, line 251 use of English
Page 8, line 255 “and” instead of “und”
Author Response
1) The main result is the proximity of the immune cells to the CD34+ cells in marrow samples of patients as compared to controls. In the materials and methods section, they state that “CD34+ blast cells were separated from CD34+ endothelial cells by histomorphology based on their localization and cytological appearance”. This appears not enough to exclude that the vicinity of the immune cells to CD34+ ones by multispectral imaging relies on endothelial proximity (rather than on blast cells). If we assume that the “histomorphologic and cytologic” criterion is enough, the same could be stated for the identification of the immune cells and the strength of the study design would be further weakened.
Thank you very much for your valuable comments. Indeed, the message of our paper was not clearly exposed. Therefore we rewrote the abstract as well as main parts of the whole manuscript. Furthermore, we discussed the strength and weakness of our study including the method employed.
The staining pattern using a panel of six antibodies, which stained different lymphocyte subsets and in addition including CD34+ cells, was independently analyzed by two hematopathologists (MB, CW). To exclude CD34+endothelial cells and include CD34+ HSPC/blasts, a visual discrimination was performed. As demonstrated in Figure 1 and described in the results part, endothelial cells are characterized by a line-shaped luminal CD34 positivity that in fact can be distinguished from CD34+ blasts displaying a complete membranous staining. As an evidence for accurate discrimination, the lack of MUM1p+ plasma cells in the control BMB is postulated, since these cells are typically found in close relationship to small vascular structures within the BM.
2) The study does not answer, not even discuss, the question whether the immune cells establish a functional relationship/signaling/communication with the MDS stem cells: were electronic microscopy experiments performed? (even in the literature), are microparticles or other mediators implied? Some functional studies would be more necessary to clarify this issue.
Despite further studies are needed to clarify the functional relevance of our findings, these are beyond of the scope of this article. However, this issue was included into the discussion.
3) The Authors may consider enriching the number of markers used to better identify the various immune cell populations (i.e. CD4, CD16, CD56, CD25, CD19, CD38 etc...) since all of them have different functional significances. If an extended evaluation is not performed, the functional role of immune cells close to CD34+ HSCs (if not vascular cells) remains unclear.
We would like to thank the reviewer for this valuable comment. We are aware that we used only a limited number of Abs, which allowed to distinguish lymphocyte subpopulations in addition to CD34+ cells. Thus, more stainings targeting additional immune cell populations as well as the immune regulation and functional status of different immune cell subsets are currently performed in order to get better insights into the different cellular players of the normal and neoplastic BM micromilieu. In this article, we focus on the main lymphocytic subsets of the adaptive immune system, which have not yet been analyzed regarding their frequency and spatial distribution to CD34+ blasts and HSCP.
4) Table 1 contains scanty clinical information and may be implemented by adding positivity of autoimmunity markers, and death. Moreover, abbreviations should be better defined.
As suggested, we added the number of patients with known autoimmune disorders and patients that died in the course of the disease. Furthermore, the abbreviations were defined.
5) The numbers of “immune” cells in treated and untreated patients are given as “average” with no range, and is often < to 0. This Referee feels that it would be better to give values ad median (range) and to define the number (%) of patients/samples that displayed higher frequency of “immune cell populations”.
Thanks for this comment. As suggested, we added the range of the numbers and summarized the data in supplementary Table 1.
6) Overall, the results are too long and not clearly exposed. I would suggest to improve figures (i.e. 3B and 3C, and 4B and 4C) that are sometimes hardly readable, and to include a table where the reader can easily distinguish median number of each cell population for each distance considered (rows) for – all patients – treated – untreated – IPSS categories etc (columns).
We improved the figures as requested and added a supplementary Table 1.
7) The mutational analysis is interesting, but it is not the focus of the article and table 2 is nice but “out of the blue”. Data are discussed altogether (MDS and AML) and at the end of the paragraph the reader is confused: is there a relationship with presence and spatial distribution of “immune cells”? The association of response to therapy and somatic mutations appears out of the focus of the study and might be removed if not linked to the immune cells infiltrate.
The table was better embedded in the text: the rational of this analysis in the context of MSI was given, the relevance of some mutations such as e.g. TP53 on the immune cell infiltration is discussed.
8) The discussion appears as a small dissertation of previous evidence, without clearly showing the added value of the present study. An effort should made to build the discussion on Authors’ findings, addressing the strengths and weaknesses of the study, the future implications, and the likely pathogenic, prognostic and therapeutic impact
As suggested, the discussion was rewritten in most parts and was built on our findings. The rational of the experiments and the results were discussed in the context of the literature and also the “pros” and “cons” of the MSI to other results was given.
Page 8, line 251 use of English
This was changed as requested.
Page 8, line 255 “and” instead of “und”
The typo was changed.
Reviewer 2 Report
I have some problems with understanding the design of the study. As far as I understand patients at various stages of therapy /no treatment are mixed.
Unless I do not understand the set-up, the patients (bone marrows) following allogeneic stem cell transplantation should stand out dramatically as compared to other subgroups with regard to immune subpopulations and immune microenvironment (bacause of conditioning, immunosuppresion for GvHD prevention, and always some GvHD. I cannot see any such data.
Apart from that I cannot see translational or prognostic consequences of the findings -unfortunately there are few correlations with disease subtype of behaviour. So what could be the application.
Also, I am not convinced inflammatory disorders particularly frequently coexist with MDS, at least when age-matched.
Author Response
I have some problems with understanding the design of the study. As far as I understand patients at various stages of therapy /no treatment are mixed.
Unless I do not understand the set-up, the patients (bone marrows) following allogeneic stem cell transplantation should stand out dramatically as compared to other subgroups with regard to immune subpopulations and immune microenvironment (because of conditioning, immunosuppression for GvHD prevention, and always some GvHD. I cannot see any such data.
We described the design of the study and the selection of patients analyzed. BMB from patients with MDS were stained at the time of diagnosis. Since patients received different treatments (ASCT, HMA, supportive treatment), we excluded all patients with HMA treatment and ASCT in the course of the disease for prognosis assessment. We did not examine any BMB of patients following allogeneic stem cell transplantation. We changed the paragraph in the results as follows:
“To exclude therapy effects, BMB of patients with prior HMA treatment and BMB of patients treated with ASCT in the course of the disease were excluded as well. The remaining patients were separated into patients with (n=6) and without (n=29) progress to sAML in the course of MDS (Table 1).”
Apart from that I cannot see translational or prognostic consequences of the findings -unfortunately there are few correlations with disease subtype of behavior. So what could be the application.
Also, I am not convinced inflammatory disorders particularly frequently coexist with MDS, at least when age-matched.
In the patient cohort analyzed, a significant number of patients with MDS and/or sAML showed autoimmune disorders as shown in Table 1.
Reviewer 3 Report
The Manuscript “Altered spatial composition of the immune cell repertoire in the bone marrow stem cell niche in myelodysplastic syndromes and secondary acute myeloid leukemia” is an interesting work focused on the identification of prognostic factors in MDS and sAML, such as the frequency and the spatial distribution of immune cell subpopulations. The work is promising; however, it still requires minor modifications:
- Some images are not clear, the authors could improve their quality and increase the font size to allow better understanding for the reader.
- At line 351, disease is written incorrectly.
- The font size could be uniform throughout the entire text.
As the authors clearly describe the role of the bone marrow niche, they could also comment on recent data about that:
- Ruvolo PP. Galectins as regulators of cell survival in the leukemia niche. Adv Biol Regul. 2019 Jan;71:41-54. doi: 10.1016/j.jbior.2018.09.003. Epub 2018 Sep 12. PMID: 30245264.
Moreover, as the mutation analysis was focused on specific genes, the authors could further describe the role of the genes in MDS pathogenesis, taking inspiration from the following papers:
- Follo MY, Pellagatti A, Ratti S, Ramazzotti G, Faenza I, Fiume R, Mongiorgi S, Suh PG, McCubrey JA, Manzoli L, Boultwood J, Cocco L. Recent advances in MDS mutation landscape: Splicing and signalling. Adv Biol Regul. 2020 Jan;75:100673. doi: 10.1016/j.jbior.2019.100673. Epub 2019 Nov 5. PMID: 31711974.
- Pellagatti A, Boultwood J. Splicing factor mutant myelodysplastic syndromes: Recent advances. Adv Biol Regul. 2020 Jan;75:100655. doi: 10.1016/j.jbior.2019.100655. Epub 2019 Sep 19. PMID: 31558432.
- Bernard E, Nannya Y, Hasserjian RP, Devlin SM, Tuechler H, Medina-Martinez JS, Yoshizato T, Shiozawa Y, Saiki R, Malcovati L, Levine MF, Arango JE, Zhou Y, Solé F, Cargo CA, Haase D, Creignou M, Germing U, Zhang Y, Gundem G, Sarian A, van de Loosdrecht AA, Jädersten M, Tobiasson M, Kosmider O, Follo MY, Thol F, Pinheiro RF, Santini V, Kotsianidis I, Boultwood J, Santos FPS, Schanz J, Kasahara S, Ishikawa T, Tsurumi H, Takaori-Kondo A, Kiguchi T, Polprasert C, Bennett JM, Klimek VM, Savona MR, Belickova M, Ganster C, Palomo L, Sanz G, Ades L, Della Porta MG, Smith AG, Werner Y, Patel M, Viale A, Vanness K, Neuberg DS, Stevenson KE, Menghrajani K, Bolton KL, Fenaux P, Pellagatti A, Platzbecker U, Heuser M, Valent P, Chiba S, Miyazaki Y, Finelli C, Voso MT, Shih LY, Fontenay M, Jansen JH, Cervera J, Atsuta Y, Gattermann N, Ebert BL, Bejar R, Greenberg PL, Cazzola M, Hellström-Lindberg E, Ogawa S, Papaemmanuil E. Implications of TP53 allelic state for genome stability, clinical presentation and outcomes in myelodysplastic syndromes. Nat Med. 2020 Oct;26(10):1549-1556. doi: 10.1038/s41591-020-1008-z. Epub 2020 Aug 3. PMID: 32747829.
The Manuscript highlights clinical relevance for the prognostic impact of the tumors, analyzing both genetic and cytological characteristics. The data is significant since the patient cohort consists of a large number. The main idea at the base of this work is interesting and opportune in this field and fit well with the journal scope. Certainly, further investigations to identify immunotherapies for MDS and sAML are needed.
Author Response
The Manuscript “Altered spatial composition of the immune cell repertoire in the bone marrow stem cell niche in myelodysplastic syndromes and secondary acute myeloid leukemia” is an interesting work focused on the identification of prognostic factors in MDS and sAML, such as the frequency and the spatial distribution of immune cell subpopulations. The work is promising; however, it still requires minor modifications:
Some images are not clear, the authors could improve their quality and increase the font size to allow better understanding for the reader. .
The quality of images was improved as suggested.
At line 351, disease is written incorrectly.
The typo was changed.
The font size could be uniform throughout the entire text.
The font size is now uniform in the text as suggested.
As the authors clearly describe the role of the bone marrow niche, they could also comment on recent data about that:
We have included the suggested references in the introduction and also in the discussion. The data were also extensively discussed in the view of our results
Ruvolo PP. Galectins as regulators of cell survival in the leukemia niche. Adv Biol Regul. 2019 Jan;71:41-54. doi: 10.1016/j.jbior.2018.09.003. Epub 2018 Sep 12. PMID: 30245264.
Moreover, as the mutation analysis was focused on specific genes, the authors could further describe the role of the genes in MDS pathogenesis, taking inspiration from the following papers:
Follo MY, Pellagatti A, Ratti S, Ramazzotti G, Faenza I, Fiume R, Mongiorgi S, Suh PG, McCubrey JA, Manzoli L, Boultwood J, Cocco L. Recent advances in MDS mutation landscape: Splicing and signalling. Adv Biol Regul. 2020 Jan;75:100673. doi: 10.1016/j.jbior.2019.100673. Epub 2019 Nov 5. PMID: 31711974.
Pellagatti A, Boultwood J. Splicing factor mutant myelodysplastic syndromes: Recent advances. Adv Biol Regul. 2020 Jan;75:100655. doi: 10.1016/j.jbior.2019.100655. Epub 2019 Sep 19. PMID: 31558432. Bernard E, Nannya Y, Hasserjian RP, Devlin SM, Tuechler H, Medina-Martinez JS, Yoshizato T, Shiozawa Y, Saiki R, Malcovati L, Levine MF, Arango JE, Zhou Y, Solé F, Cargo CA, Haase D, Creignou M, Germing U, Zhang Y, Gundem G, Sarian A, van de Loosdrecht AA, Jädersten M, Tobiasson M, Kosmider O, Follo MY, Thol F, Pinheiro RF, Santini V, Kotsianidis I, Boultwood J, Santos FPS, Schanz J, Kasahara S, Ishikawa T, Tsurumi H, Takaori-Kondo A, Kiguchi T, Polprasert C, Bennett JM, Klimek VM, Savona MR, Belickova M, Ganster C, Palomo L, Sanz G, Ades L, Della Porta MG, Smith AG, Werner Y, Patel M, Viale A, Vanness K, Neuberg DS, Stevenson KE, Menghrajani K, Bolton KL, Fenaux P, Pellagatti A, Platzbecker U, Heuser M, Valent P, Chiba S, Miyazaki Y, Finelli C, Voso MT, Shih LY, Fontenay M, Jansen JH, Cervera J, Atsuta Y, Gattermann N, Ebert BL, Bejar R, Greenberg PL, Cazzola M, Hellström-Lindberg E, Ogawa S, Papaemmanuil E. Implications of TP53 allelic state for genome stability, clinical presentation and outcomes in myelodysplastic syndromes. Nat Med. 2020 Oct;26(10):1549-1556. doi: 10.1038/s41591-020-1008-z. Epub 2020 Aug 3. PMID: 32747829.
As suggested, the literature was integrated into the manuscript at the appropriate localization.
The Manuscript highlights clinical relevance for the prognostic impact of the tumors, analyzing both genetic and cytological characteristics. The data is significant since the patient cohort consists of a large number. The main idea at the base of this work is interesting and opportune in this field and fit well with the journal scope. Certainly, further investigations to identify immunotherapies for MDS and sAML are needed.
Round 2
Reviewer 2 Report
I still see little translational application, so this can be accepted as a preliminary study -openning a topic for further research. This should be elaborated on.
There are some corralations, but there is no clue whether the differences are causative for MDS of consequences of bone marrow alteration. Any ideas from the authors?
There is no attempt at correlation this spatial immune alterations with any other features discribing the immune status of the patients, such as lymphocyte populations outside bone marrow niche.
The authors should provided the types of autoimmunity (n=10) seen in MDS patients. E.g. diabetes or Hashimoto is very far away from rheumatoid arthritis or lups. How were the controls examined for autoimmunity?
One more ctiticism is that the contrlols are circa 10 years younger (median age?) The may impact on autoimmunity and bone marrow immune spatial composition. I understand the groups are not probably different statictically, but there was also no power to detect difference in age.
Author Response
I still see little translational application, so this can be accepted as a preliminary study -openning a topic for further research. This should be elaborated on.
There are some corralations, but there is no clue whether the differences are causative for MDS of consequences of bone marrow alteration. Any ideas from the authors?
Thank you very much for your valuable comments. Indeed, we expected different results with a correlation between the immune cell infiltration, the genetics and the course of the disease. Additional investigations have to be performed to get more insights into the heterogeneity of the immune cell infiltrate in spatial proximity of CD34+ blasts in MDS and sAML samples that might help to understand the relationship of the disease and the bone marrow alterations including the composition of the immune subpopulations. This might have an impact on diagnosis and prognostics of MDS/AML, but also on therapeutic strategies.
There is no attempt at correlation this spatial immune alterations with any other features discribing the immune status of the patients, such as lymphocyte populations outside bone marrow niche.
Thank you very much for this comment. Indeed, we have now analyzed whether there exists a correlation between the leukocyte counts of the peripheral blood and the immune cell frequency and the spatial distribution of the immune cell subsets in the BM. However, MDS patients with elevated peripheral blood leukocyte counts exhibited slightly increased numbers of MUM1p+CD3- post-germinal center B/plasma cells (p=0.058) and CD3+ T cells (p=0.061). There is no statistically significant correlation between the peripheral leukocyte counts and the spatial distribution of the immune cell subsets in the BM. Furthermore, we added the leukocyte counts of the peripheral blood in Table 1.
The authors should provided the types of autoimmunity (n=10) seen in MDS patients. E.g. diabetes or Hashimoto is very far away from rheumatoid arthritis or lups. How were the controls examined for autoimmunity?
The patients with known immune disorders suffered from autoimmune thyroiditis (Hashimoto thyroiditis or Graves’ disease), autoimmune cholangitis, myasthenia gravis or rheumatoid arthritis. This information was added as types of auto(-immune) disorders in Table 1.
One more ctiticism is that the contrlols are circa 10 years younger (median age?) The may impact on autoimmunity and bone marrow immune spatial composition. I understand the groups are not probably different statictically, but there was also no power to detect difference in age.
The statement of the reviewer is right. Despite the controls are in average about ten years younger compared to the MDS and sAML patients, the range of the patients’ age is comparable and there are also very old controls (>80 years).